# 📦 $\mathbb{T}^3$: Test-Time Model Merging in VLMs for Zero-Shot Medical Imaging Analysis

## Abstract

In medical imaging, vision-language models face a critical duality: *pretrained* networks offer broad robustness but lack subtle, modality-specific characteristics, while fine-tuned *expert* models achieve high in-distribution accuracy yet falter under modality shift. Existing model-merging techniques, designed for natural-image benchmarks, are simple and efficient but fail to deliver consistent gains across diverse medical modalities; their static interpolation limits reliability in varied clinical tasks. To address this, we introduce **T**est-**T**ime **T**ask adaptive merging ($\mathbb{T}^3$), a backpropagation-free framework that computes *per-sample* interpolation coefficients via the Jensen-Shannon divergence between the two models' output distributions. $\mathbb{T}^3$ dynamically preserves local precision when models agree and defers to generalist robustness under drift. To overcome the inference costs of sample-wise merging, we further propose a batch-wise extension, $\mathbb{T}^3_{\mathcal{B}}$ that computes merging coefficient across a batch of samples, dramatically reducing computational bottleneck. Recognizing the lack of a standardized medical-merging benchmark, we present a rigorous cross-evaluation protocol spanning in-domain, base-to-novel, and corruptions across four modalities. Empirically, $\mathbb{T}^3$ sets new state-of-the-art in Top-1 accuracy and error reduction, outperforming strong baselines while maintaining efficiency, paving the way for adaptive MVLM deployment in clinical settings.

## 1 Introduction

Medical vision-language models (MVLMs) are typically developed in two flavors: (1) expert models obtained via fine-tuning on domain-specific data, which are highly specialized but may overfit to in-distribution cases, and (2) pretrained models that provide strong generalization but may lack the domain-specific nuances. In healthcare settings, we imagine having two "clinicians" on call: a local specialist, expert MVLM, whose expertise is honed on a single hospital's scanner, patient demographics, and imaging protocols, and a global generalist, a large-scale pretrained MVLM, whose broad training spans many sites, scanners, and pathologies. The specialist delivers higher confidence in familiar cases but may falter when faced with a case that varies from the norm, e.g., a medical scan from a new device or an unseen patient population. The generalist is typically robust to such distribution shifts but may lack the fine-grained, site-specific nuance, e.g., a specific hospital's imaging protocols. *Clinicians typically take advice from each other, especially on challenging cases, e.g., a neurologist may consult a radiologist on his/her views of a patient's brain atrophy appearing on MRI scans to be able to diagnose neurodegenerative diseases. This form of multiple views decision making could be mimicked by machine learning models using a simple voting mechanism or more sophisticated model merging methods* Wortsman et al. (2022b;a).

Existing model-merging strategies typically choose a fixed blend of these two models or rely on simple heuristics that cannot distinguish when the specialist's local insight truly applies versus when the generalist's broad knowledge should take precedence. This leaves a critical gap: how to efficiently fuse specialist and generalist decisions to reach to an accurate clinical outcome by utilizing the right aggregation of their knowledge. While prior works in model merging Wortsman et al. (2022b) has focused on fixed or globally optimized merging weights, there is limited exploration into dynamic,

Table 1: Comparison of Static and Dynamic merging methods along three critical dimensions: practicality, domain generalizability, and test-time adaptability. ● indicates that the method exhibits the desired trait for that criterion, whereas ○ indicates that it does not. $\mathbb{T}^3$ excels in all dimensions. Given a pretrained and a expert model, inference cost (I) is measured in forward-passes over the entire test set (with $N$ total samples, grouped into $B$ batches of size BS so that $N = B \times$ BS, where $N >> B$). See Section 5.2 for details. Accuracy and Robustness indicates Top-1 Acc and Err (See Section 4) averaged over *mean OOD* of 4 medical datasets.

| Method | Venue | Practical (No Training) | Universal (Modality-Generalizable) | TTA (Adaptive) | Consistency (Accuracy ↑) | Robustness (Error ↓) | Cost (#Forwards) |
|---|---|---|---|---|---|---|---|
| Pretrained CLIP | - | - | ● | ○ | 38.05 | 100.0 | $\mathcal{O}(1B)$ |
| Expert CLIP | - | - | ○ | ○ | 55.01 | 86.7 | $\mathcal{O}(1B)$ |
| Model Ensemble | - | ● | ● | ○ | 33.70 | 111.3 | $\mathcal{O}(2B)$ |
| Model Souping | PMLR'24 | ● | ● | ○ | 41.82 | 93.5 | $\mathcal{O}(1B)$ |
| Task Arithmetic | ICLR'23 | ● | ○ | ○ | 44.77 | 99.4 | $\mathcal{O}(1B)$ |
| Slerp | NIPS'16 | ● | ○ | ○ | 41.82 | 93.5 | $\mathcal{O}(1B)$ |
| Mixup Merging | arXiv'25 | ● | ○ | ○ | 49.82 | 84.1 | $\mathcal{O}(1B)$ |
| DaWin | ICLR'25 | ● | ○ | ● | 44.48 | 89.0 | $\mathcal{O}(3B)$ |
| Sample-wise Merge | Ours $\mathbb{T}^3$ | ● | ● | ● | 58.05 | 71.9 | $\mathcal{O}(N)$ |
| Batch-wise Merge | Ours $\mathbb{T}^3_B$ | ● | ● | ● | 58.17 | 71.7 | $\mathcal{O}(1B)$ |

sample- or batch-wise adaptation strategies for merging expert[1] and pretrained models. For instance, Wise-FT Wortsman et al. (2022b) merge these models using an average interpolation factor (*i.e.*, $\alpha$=0.5), yet this does not account for the variability in test samples or batches, which may exhibit different levels of domain shift. An integrated solution is required to balance the trade-off between generalization and specialization in VLMs while enhancing zero-shot generalization during inference for a single test sample or a batch.

Therefore, weight-interpolation methods Wortsman et al. (2022b); Lu et al. (2024); Oh et al. (2024); Yang et al. (2023) remain untested in MVLMs under diagnostic test-time conditions. Medical imaging presents high inter-patient variability, protocol differences, and scarce annotations, calling for a flexible, training-free adaptation rule that balances expert specialization with pretrained robustness. Moreover, without a standardized evaluation protocol, it is difficult to gauge how merging strategies handle domain shifts across medical modalities. In clinical diagnostics, reliable performance across diverse datasets is paramount, far outweighing marginal improvements on any single modality. These gaps motivate our core research questions:

> **Research Questions**
>
> 1. In what ways can the synergistic performance of pretrained and domain-expert MVLMs be maximized, *consistently* for various medical tasks?
> 2. How can pretrained and expert MVLMs balance *consensus and divergence* in the predicted probability distribution for a given test input?
> 3. Under conditions of distribution shifts, to what extent do static and dynamic merging methods demonstrate robustness and maintain *consistent* generalizability across medical modalities?

Our proposed $\mathbb{T}^3$ framework is motivated by the need to bridge the gap between specialized and general models when dealing with OOD samples in medical imaging. By learning an adaptive interpolation weight for each sample or batch at test time, $\mathbb{T}^3$ dynamically modulates the contributions of the expert and pretrained models. Additionally, the combination of corruption medical datasets (which simulate noise and artifacts) with novel class datasets provides an excellent out-of-distribution benchmark for testing model merging approaches. This comprehensive approach allows us to rigorously evaluate and improve model merging performance in both degraded and unseen conditions. Our contributions can be summarized as follows:

– We introduce $\mathbb{T}^3$ (pronounced /tee:cube/), a non-iterative and backpropagation-free **Test-Time Task** adaptive interpolation framework that learns optimal batch-wise merging weights without incurring the high computational cost of full backpropagation.

– We establish a **benchmark** for model merging in medical imaging by proposing a hard yet practical cross-evaluation protocol ranging various medical OOD scenarios, assessing across corrupted inputs (MedMNIST-C Di Salvo et al. (2024)) and novel class generalization (MediMeta Woerner et al. (2024b)).

---

[1]We use "domain expert" and "modality expert" interchangeably, where "domain" denotes the data distribution of a specific medical modality.

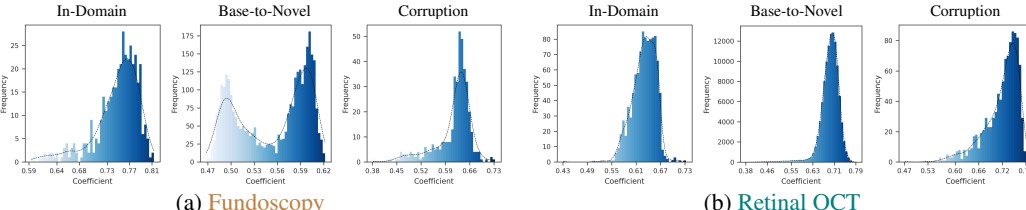

(a) Fundoscopy          (b) Retinal OCT

Figure 1: **Histogram of interpolation coefficients induced by X-entropy ratio $X(x)$ (from Eq. 4) between pretrained and expert models.** For each modality and under three test settings: In-Domain (data seen to *expert* during fine-tuning), Base-to-Novel (cross-dataset generalization), and Corruption inputs. This shows that $X(x)$ coefficient estimates vary greatly and is dependent on different data modality and OOD shifts regarding symmetry and skewness. For instance, in Fundoscopy, $X(x)$ remains tightly clustered for In-Domain testset but shows strong variation under Base-to-Novel inputs, indicating reduced reliance on the fine-tuned expert.

– We provide an **empirical analysis** that justifies the test-time dynamic model merging and its benefits in addressing distribution shifts consistently across multiple medical tasks.

– We demonstrate that $\mathbb{T}^3$ robustly outperforms fine-tuned models in an unsupervised manner consistently across four medical modality tasks, setting **state-of-the-art** results on the proposed benchmark for model merging in medical VLMs[2].

## 2 RELATED WORKS

**Test-Time Adaptation in VLMs:** To address *distribution shifts* and improve OOD generalization, recent works have explored Test-Time Adaptation (TTA) techniques Shu et al. (2022); Abdul Samadh et al. (2023); Feng et al. (2023); Zanella & Ben Ayed (2024); Imam et al. (2024). Given an input image and a set of class descriptions, TTA typically involves a trainable component, such as a learnable prompt, optimized using an entropy-based objective function derived from multiple augmentations of the test sample. The adapted component is then used for final inference. However, existing TTA methods often require multiple augmentations and optimization. To overcome these limitations, we aim to propose a *backpropagation-free* merging method that enhances efficiency and reduces memory overhead, making it more suitable for resource-constrained clinical environments.

**Model Merging:** In the context of medical imaging, model merging is particularly valuable for adapting to diverse and noisy clinical scenarios, where maintaining a balance between specialized and generalizable representations is crucial Wortsman et al. (2022b). Existing works on model merging, such as AdaMerging Yang et al. (2023), TiesMerging Yadav et al. (2023), and Model Soups Wortsman et al. (2022a) have primarily focused on natural image distributions, while Wang et al. (2025) focused on medical distributions but applied to relatively smaller CNN architectures. These methods often lack the adaptive mechanisms required to effectively merge weights across varying corruption scenarios in medical data. Thus, our $\mathbb{T}^3$ framework is designed to bridge the gap between test-time adaptation in large-scale MVLMs and model merging strategies in medical imaging[3].

## 3 METHODOLOGY

### 3.1 PROBLEM SETUP

We consider a $C$-way classification task over a test set $\mathcal{D}_{\text{test}} = \{x_i\}_{i=1}^N$, where each input $x \in \mathcal{X}$ must be assigned one of $C$ discrete class labels. Let $f_{\text{pt}}$ and $f_{\text{ft}}$ denote the pretrained and finetuned MVLMs, respectively, each comprising an image-text encoder architecture similar to CLIP, adapted for a common medical modality (e.g., cell classification). For each test input $x \in \mathcal{D}_{\text{test}}$, the image-text encoder processes $x$ alongside class-level textual prompts to compute the similarity scores, followed by then converting into logits $z_{\text{pt}}(x)$, $z_{\text{ft}}(x) \in \mathbb{R}^C$, where $C$ is the number of classes. The corresponding softmax outputs,

$$p_{\text{pt}}(x) = \text{softmax}\big(z_{\text{pt}}(x)\big) \quad \text{and} \quad p_{\text{ft}}(x) = \text{softmax}\big(z_{\text{ft}}(x)\big), \qquad (1)$$

---

[2]We will release our codebase and benchmarking setup upon paper acceptance.

[3]Additional Related Works are discussed in Appendix B.

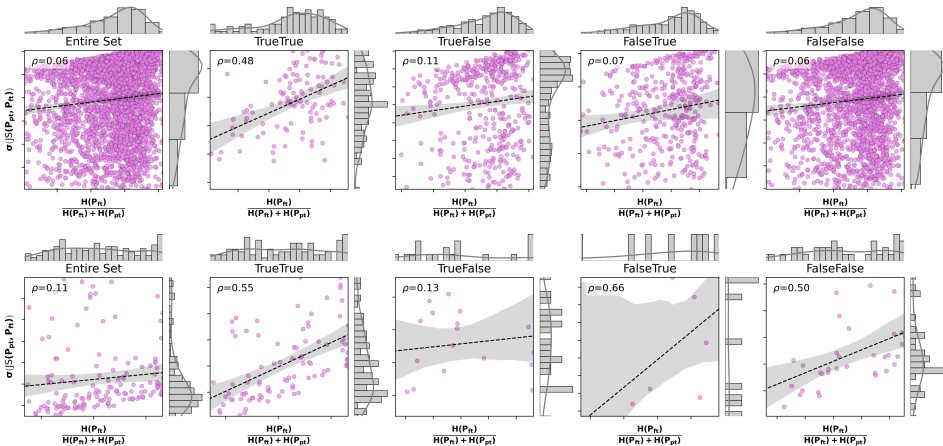

Figure 2: **Pearson correlation $\rho$ between Mutual Information $I(x)$ (Eq. 5)** *and* **Entropy-ratio $R(x)$ (Eq. 3).** We partition each test set into four groups—TrueTrue, TrueFalse, FalseTrue, and FalseFalse—according to whether the Pretrained and Expert models make correct or incorrect predictions. For each group, we plot Pearson correlation $\rho$ scatter of the entropy ratio $R(x)$ on the x-axis against the Mutual Information $I(x)$. Top row denotes Cell Microscopy PBC (from MediMeta) dataset while Bottom row denotes Breast Imaging Mammo MediMeta dataset with CLIP ViT-B/16 backbone. This correlation implies that $I(x)$ strongly correlates with the $R(x)$ overall across all groups, suggesting a strong alternative interpolation coefficient that could also capture joint predictive confidence better than entropy.

define the confidence distributions over the prompt concatenated class labels. Our goal is to design a *test-time merging* procedure that, for each $x \in \mathcal{D}_{\text{test}}$, computes a sample-specific interpolation coefficient $\lambda(x) \in [\lambda_{\min}, \lambda_{\max}]$ and then fuses the two parameter sets as

$$\theta_{\text{merged}}(x) = (1 - \lambda(x))\,\theta_{\text{pt}} + \lambda(x)\,\theta_{\text{ft}}. \tag{2}$$

The resulting model $f_{\text{merged}}(\,\cdot\,;\,\theta_{\text{merged}}(x))$ is thus a data-dependent convex combination of the generic (pretrained) and specialized (fine-tuned) hypotheses.

> **Analogy of Merging Setup in Diagnostics**
>
> Think of it like having two radiologists on call: one's a local specialist tuned to this hospital's scanner and patient population (our fine-tuned MVLM), and the other's a global expert with broad experience across many scanners and cohorts (our pretrained MVLM). When a tricky new scan arrives, outside the specialist's usual cases or exhibiting unfamiliar artifacts, we compute a per-scan mixing weight $\lambda(x)$ that blends the specialist's local insight with the expert's wide-ranging knowledge. The result is a hybrid diagnosis that adapts sample by sample, giving us the specialist's precision when it's reliable and the generalist's robustness when it's needed (See Figure 7 in Appendix).

### 3.2 DESIGNING DYNAMIC COEFFICIENT

We hypothesize that measuring mutual information, e.g. via Jensen-Shannon divergence, between a pretrained model's output and its fine-tuned counterpart offers a more faithful gauge of their joint predictive confidence than simply combining their entropies, since it explicitly distinguishes when the two models agree versus disagree. For comparison, DaWin Oh et al. (2024) introduces an *entropy ratio*

$$R(x) = \mathcal{H}(p_{\text{ft}}(x))/[\mathcal{H}(p_{\text{pt}}(x)) + \mathcal{H}(p_{\text{ft}}(x))] \tag{3}$$

where $p_{\text{pt}}(x)$ and $p_{\text{ft}}(x)$ are the predictive distributions on input $x$ (as from Eq. 1), and $H(\cdot)$ denotes self-entropy. This ratio effectively interpolates between the two uncertainties. While combined confidence in $R(x)$ simply averages each model's top-1 prediction into a single scalar, JS divergence instead measures the relationship between their full predictive distributions and thus more directly flags high-confidence disagreements that a top-class aggregation alone would miss as empirically evident in Figure 3. Across, natural image tasks, DaWin shows that $R(x)$ correlates positively with the *cross-entropy ratio* or *X-entropy ratio*

$$X(x) = \ell(p_{\text{ft}}(x), y)/[\ell(p_{\text{pt}}(x), y) + \ell(p_{\text{ft}}(x), y)], \tag{4}$$

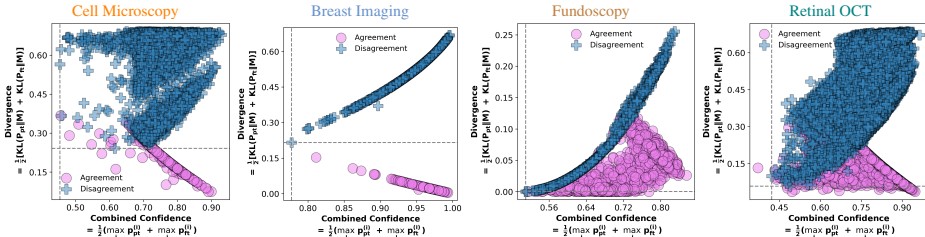

Figure 3: **Decision-Quadrant Analysis of *Consensus vs. Disagreement* via Combined Confidence and JS Divergence**. Here **M** refers to $\bar{p}(x)$ as in Eq. 5. While combined confidence alone treats high-confidence OOD samples uniformly—failing to separate agreement from disagreement—JS divergence cleanly isolates high-confidence disagreements, highlighting its superiority as a proxy for joint predictive certainty in model-merging scenarios across diverse modalities.

where $\ell(p, y)$ is the cross-entropy loss of distribution $p$ with true label $y$. A high $R(x)$ generally indicates that the fine-tuned model is more certain (lower loss) than the pretrained model, which empirically aligns with better predictive accuracy. However, $R(x)$ can be misleading when both models are confident but *disagree*: if $p_{\text{pt}}$ and $p_{\text{ft}}$ are both sharp (low entropy) but confident on different classes, the ratio $R(x)$ gives no indication of this conflict. To capture such "consensus versus disagreement", we introduce the mutual information $I(x)$ between the two output distributions (See Figure 3). Concretely, let $\bar{p}(x) = \frac{1}{2}\big(p_{\text{pt}}(x) + p_{\text{ft}}(x)\big)$ be the average distribution. We define

$$I(x) = \frac{1}{2}\Big(\text{KL}(p_{\text{pt}}(x)\|\bar{p}(x)) + \text{KL}(p_{\text{ft}}(x)\|\bar{p}(x))\Big), \tag{5}$$

which is exactly the Jensen-Shannon divergence between $p_{\text{pt}}(x)$ and $p_{\text{ft}}(x)$. Equivalently, using the convexity of entropy,

$$I(x) = H\big(\bar{p}(x)\big) - \frac{1}{2}\big(H(p_{\text{pt}}(x)) + H(p_{\text{ft}}(x))\big). \tag{6}$$

This formulation has several desirable properties. If the two models *fully agree* on their predictions, then $p_{\text{pt}}(x) = p_{\text{ft}}(x)$ and $I(x) = 0$. Conversely, if they are both confident but assign high probability to different classes, then $\text{KL}(p_{\text{pt}}\|\bar{p})$ and $\text{KL}(p_{\text{ft}}\|\bar{p})$ are large, so $I(x)$ becomes large. In effect, $I(x)$ quantifies the "consensus-disagreement" structure: it remains low when models agree (even if confident) and increases when they disagree (even if each is confident). Empirically, Figure 2 observes a *consistently* positive correlation between $I(x)$ and $R(x)$,

$$\text{Corr}\big(I(x), R(x)\big) > 0, \tag{7}$$

i.e., inputs with high $R(x)$ ($f_{\text{ft}}$ more confident than $f_{\text{pt}}$) tend to also have high $I(x)$, but importantly $I(x)$ can further distinguish cases of disagreement that $R(x)$ alone would miss. This suggests that mutual information indeed captures joint predictive confidence more faithfully than entropy ratio alone, validating our design choice.

### 3.3 $\mathbb{T}^3$ MERGING WORKFLOW

**Objective:** We consider two fixed classifiers: a pretrained model $f_{\text{pt}}$ with parameters $\theta_{\text{pt}}$, and a fine-tuned model $f_{\text{ft}}$ with parameters $\theta_{\text{ft}}$. For an input $x$, these produce output distributions $p_{\text{pt}}(x)$ and $p_{\text{ft}}(x)$ (e.g. softmax probabilities). Our goal is to *adaptively merge* the two models at test time by forming a weighted average of their parameters. Concretely, we define a sample-wise merged model

$$\theta_{\text{merged}}(x) = (1 - \lambda(x))\,\theta_{\text{pt}} + \lambda(x)\,\theta_{\text{ft}}, \tag{8}$$

where the interpolation coefficient $\lambda(x)$ depends on $x$. If $\lambda(x) = 0$, the merged model is just the pretrained network; if $\lambda(x) = 1$, it is the fine-tuned network. In general, $\lambda(x) \in [\lambda_{\min}, \lambda_{\max}]$ balances the two. Crucially, this merging is performed in a test-time, unsupervised manner: no ground-truth labels are available during merging. In practice, $\theta_{\text{merged}}$ is computed on-the-fly from the two models' outputs, with $\lambda(x)$ determined by our Jensen-Shannon criterion. This operation is model-agnostic and requires only simple post-processing of the outputs, making it straightforward to integrate into existing inference pipelines as shown in Figure 4.

Figure 4: $\mathbb{T}^3$ **Test-Time Task Adaptive Merging Workflow.** For each input $x$, both pretrained CLIP and domain expert models generate output distributions that are compared using Jensen-Shannon divergence to quantify their agreement. This divergence is transformed into an interpolation coefficient $\lambda(x)$ through sigmoid function, which determines the specific parameter blending for each *test* sample. Higher disagreement (larger JS divergence) increases the expert model's influence, while agreement favors the pretrained model, enabling adaptive merging that optimizes both accuracy and robustness across distribution shifts.

**Mutual Information-Guided Interpolation:** To choose $\lambda(x)$ for each input, we quantify the *agreement* between the two model predictions via the Jensen-Shannon (JS) divergence. Specifically, we define a per-input Mutual Information (MI) score

$$I(x) \;=\; \mathrm{JS}\big(p_{\mathrm{pt}}(x),\, p_{\mathrm{ft}}(x)\big) \;=\; \tfrac{1}{2}\Big[\mathrm{KL}\big(p_{\mathrm{pt}}(x)\,\|\,\bar{p}(x)\big) + \mathrm{KL}\big(p_{\mathrm{ft}}(x)\,\|\,\bar{p}(x)\big)\Big], \qquad (9)$$

where $\bar{p}(x) = \tfrac{1}{2}(p_{\mathrm{pt}}(x) + p_{\mathrm{ft}}(x))$ is the mixture distribution. By construction $I(x) = 0$ if the two distributions are identical, and grows larger when they disagree. We then transform $I(x)$ into an interpolation coefficient $\lambda(x)$ via a sigmoid, ensuring a smooth, monotonic dependence on the disagreement:

$$\lambda(x) \;=\; \lambda_{\min} \;+\; (\lambda_{\max} - \lambda_{\min})\, \sigma\big(I(x)\big), \qquad (10)$$

where $\sigma(z) = 1/(1 + e^{-z})$ is the logistic sigmoid. Intuitively, this means that when the two models *agree strongly* (small $I(x)$), $\lambda(x)$ stays near $\lambda_{\min}$, and when they *disagree strongly* (large $I(x)$), $\lambda(x)$ approaches $\lambda_{\max}$. In practice we often set $\lambda_{\min} = 0$ and $\lambda_{\max} = 1$ so that $\lambda(x) \in [0, 1]$, but these bounds can be tuned. In summary, higher JS divergence drives the merged model to favor the fine-tuned parameters, whereas low divergence keeps it close to the pretrained model. This strategy provides a principled, information-theoretic way to interpolate the two networks based on how differently they "view" each input.

**Extrapolation for Extreme Confidence:** In practice, extreme confidence from one model can lead to overly aggressive interpolation weights. To address this, we introduce a small extrapolation factor $\delta > 0$ and entropy thresholds $\tau_{\mathrm{pt}}, \tau_{\mathrm{ft}}$ for the pretrained and fine-tuned models, respectively. When $f_{\mathrm{ft}}$ is exceptionally confident ($\mathcal{H}_{\mathrm{ft}}(x) < \tau_{\mathrm{ft}}$), we gently boost its influence, and when $f_{\mathrm{pt}}$ is exceptionally confident ($\mathcal{H}_{\mathrm{pt}}(x) < \tau_{\mathrm{pt}}$), we correspondingly reduce the fine-tuned weight:

$$\lambda'(x) = \begin{cases} \min\big(\lambda(x) + \delta,\, 1\big), & \mathcal{H}_{\mathrm{ft}}(x) < \tau_{\mathrm{ft}}, \\ \max\big(\lambda(x) - \delta,\, 0\big), & \mathcal{H}_{\mathrm{pt}}(x) < \tau_{\mathrm{pt}}, \\ \lambda(x), & \text{otherwise.} \end{cases} \qquad (11)$$

This conditional adjustment, clamped back into $[0, 1]$, ensures that when one model's predictive entropy is abnormally low, the merged weight is nudged toward that model, *mirroring how a clinician might rely more heavily on an imaging modality that exhibits unusually clear contrast in a given case.*

**Batch-wise Efficient Interpolation:** Naïvely, computing a unique merged model for each of the $N$ test samples would require $N$ separate parameter interpolations and forward passes, an impractical cost in large-scale deployment. To alleviate this burden, we instead partition the $N$ inputs into $B$ disjoint batches $\{\mathcal{B}_b\}_{b=1}^{B}$. For each batch $\mathcal{B}_b$, we compute the mean of the extrapolated interpolation weights:

$$\bar{\lambda}_b \;=\; \frac{1}{|\mathcal{B}_b|} \sum_{x_i \in \mathcal{B}_b} \lambda'(x_i), \qquad (12)$$

where $\lambda'(x)$ is defined in Eq. (11). We then perform a single merge per batch as:

$$\theta_{\mathrm{merged}}^{(b)} \;=\; (1 - \bar{\lambda}_b)\, \theta_{\mathrm{pt}} \;+\; \bar{\lambda}_b\, \theta_{\mathrm{ft}}. \qquad (13)$$

By reducing the number of distinct parameter interpolations from $N$ to $B$, we retain the sample-adaptive spirit of our MI-guided merging while cutting inference overhead by a factor of $\frac{N}{B}$. This simple batched averaging of $\lambda$ coefficients proves both efficient and effective in practice, delivering near-sample-wise performance at a fraction of the computational cost.

Figure 5: **Cross-Dataset Evaluation Benchmark**, depicting In-domain and cross-domain setup for model merging in medical imaging. This illustrates four test conditions: (i) in-domain MedMNIST Chen et al. (2021), (ii) novel-class samples from MediMeta Woerner et al. (2024b), (iii) noise corruptions (MedMNIST-C Di Salvo et al. (2024)), and (iv) pixelation corruptions (MedMNIST-C Di Salvo et al. (2024)), for each of the four imaging modalities. See Appendix 4 for details.

## 4 DATASETS AND EXPERIMENTS

**Cross-Dataset Settings:** For in-domain evaluation, we fine-tune CLIP Chen et al. (2021) on the MedMNIST Chen et al. (2021) split corresponding to its modality (e.g. RetinaMNIST for fundoscopy, BreastMNIST for breast imaging, etc.), representing the "single-hospital" data distribution that the *expert* model has seen, as also shown in Figure 5. Our choice of MedMNIST as the in-domain dataset stems from the practical observation that hospitals often have their own specific *distribution* and *quality* of in-domain data, which may differ significantly from data encountered at test time from other hospitals. A practical anology is also illustrated in Figure 7.

To probe out-of-distribution (OOD) performance, we then challenge the model with two kinds of distribution shifts: (1) a *base-to-novel* (B2N) classification task drawn from MediMeta Woerner et al. (2024a), which uses the same imaging modality but from different institutions or patient populations, and (2) medically realistic corruptions of the MedMNIST test images (from MedMNIST-C Di Salvo et al. (2024)) (noise and digital pixelation). Together, these two conditions capture both semantic shifts (new classes, new sources) and low-level perturbations (acquisition artifacts), allowing us to simulate how a model trained on one hospital's input would fare when deployed on *data from other clinics* or *under degraded imaging conditions*. Extended details that led us to formulate the aforementioned cross-dataset settings, along with **Metrics**, **Baseline** choices, and **Additional Implementation** details, are discussed in Appendix D.

> **Practicality of Medical Cross-Evaluation Protocol**
>
> In a real-world analogy, this evaluation scenario is akin to deploying a diagnostic AI that was trained on images from Hospital A, and now must interpret both unfamiliar patient cohorts at Hospital B and scans taken under suboptimal imaging settings. Detailed use-case is illustrated in Figure 7.

**Implementation Details:** All of our experiments are implemented in PyTorch and run on an NVIDIA A6000 48GB GPU. For the pretrained model, we use CLIP checkpoints across ViT-B/16 and ViT-L/14 backbones. For expert models (with *homogeneous* architecture as pretrained CLIP) across four medical modalities, we fine-tune each on the respective MedMNIST in-domain training split, attaining 4 *experts* respective to 4 modalities. At test time, for each image $x$, we compute the Jensen-Shannon divergence $I(x)$ Menéndez et al. (1997) between the pretrained and fine-tuned output distributions, and map it via a scaled sigmoid (clamped to $[\lambda_{\min} = 0.0, \ \lambda_{\max} = 1.0]$) to obtain a per-sample merging weight $\lambda(x)$ (the $\mathbb{T}^3$ variant Eq. 10). We also evaluate a batch-wise variant, $\mathbb{T}^3_{\mathcal{B}}$, in which the $N$ test samples are split into $B$ batches wit batch size $\mathrm{BS} = 32$ for efficiency. The per-sample weights $\{\lambda(x)\}$ are averaged within each batch to yield $\bar{\lambda}_{\mathcal{B}}$, which is used to perform a single parameter merge per batch (Eq. 12). To guard against overly aggressive weighting, we apply a small extrapolation step ($\delta = 0.5$) when model entropies fall below the threshold $\tau = 0.05$. We report results averaged over three runs using different random seeds to ensure robustness and reproducibility.

## 5 RESULTS AND DISCUSSION

### 5.1 MAIN RESULTS

**Accuracy *vs* Robustness:** While $\mathbb{T}^3_{\mathcal{B}}$ differs from $\mathbb{T}^3$ in terms of inference overhead, both of the proposed dynamic merging methods, consistently outperform static merging and dynamic merging

Table 2: Comparison of Top-1 Accuracy for In-Domain and Distribution shifts $\epsilon$ {Base-to-Novel (B2N), Corruption settings} on CLIP ViT-B/16 across **four** modalities. "In-Domain" refers to in-distribution data (seen to *Expert* fine-tuned CLIP) from MedMNIST, Base-to-Novel (B2N) from MediMeta, and Corruptions from MedMNIST-C. *mean* indicates the average accuracy across Distribution shifts. **Bold** highlights best performance, while underlined denotes second-best performance. Details on Baseline selection is discussed in Appendix D.

| Cell Microscopy → | In-Domain | B2N | Corruptions | | *mean* |
| Methods ↓ | BloodMNIST | PBC | Noise | Digital | mean |
| --- | --- | --- | --- | --- | --- |
| Pretrained | 16.16 | 13.73 | 16.05 | 10.14 | 13.31 |
| Expert | 98.68 | 31.21 | 88.07 | 64.40 | 61.23 |
| *Static Merging* | | | | | |
| Model Ensemble | 14.70 | 12.49 | 15.35 | 12.80 | 13.55 |
| Model Souping | 24.23 | 7.25 | 19.47 | 19.47 | 15.40 |
| Task Arithmetic | 56.68 | 14.47 | 51.97 | 21.46 | 29.30 |
| Slerp | 24.26 | 7.22 | 19.47 | 19.47 | 15.39 |
| Ties Merging | 68.69 | 4.05 | 27.65 | 31.80 | 21.17 |
| Mixup Merging | 98.71 | 31.23 | 31.37 | 66.41 | 43.00 |
| *Dynamic Merging* | | | | | |
| DaWin | 16.87 | 13.77 | 17.10 | 11.58 | 14.15 |
| $\mathbb{T}^3$ (Ours) | 98.54 | 30.68 | **86.79** | 65.24 | 60.90 |
| $\mathbb{T}^3_{\mathcal{B}}$ (Ours) | **98.66** | 31.19 | 86.29 | **66.59** | **61.36** |

| Breast Imaging → | In-Domain | B2N | Corruptions | | *mean* |
| Methods ↓ | BreastMNIST | Mammo | Noise | Digital | mean |
| --- | --- | --- | --- | --- | --- |
| Pretrained | 58.97 | 46.23 | 71.79 | 46.15 | 54.72 |
| Expert | 83.33 | 54.60 | 80.77 | 70.51 | 68.63 |
| *Static Merging* | | | | | |
| Model Ensemble | 66.67 | 53.07 | 67.95 | 50.00 | 57.01 |
| Model Souping | 78.85 | 46.23 | 73.08 | **71.79** | 63.70 |
| Task Arithmetic | 69.87 | **56.19** | 66.03 | 69.87 | 64.03 |
| Slerp | 78.85 | 46.23 | 73.08 | 71.79 | 63.70 |
| Ties Merging | 78.21 | 54.54 | 76.28 | 74.36 | 68.39 |
| Mixup Merging | 82.69 | 54.30 | 71.79 | 72.44 | 66.18 |
| *Dynamic Merging* | | | | | |
| DaWin | 71.15 | 45.99 | 77.56 | 67.95 | 63.83 |
| $\mathbb{T}^3$ (Ours) | 83.33 | 54.72 | 80.77 | 69.87 | 68.45 |
| $\mathbb{T}^3_{\mathcal{B}}$ (Ours) | 83.33 | 54.89 | 80.77 | 71.15 | 68.94 |

| Fundoscopy → | In-Domain | B2N | Corruptions | | *mean* |
| Methods ↓ | RetinaMNIST | Fundus | Noise | Digital | mean |
| --- | --- | --- | --- | --- | --- |
| Pretrained | 43.50 | 78.28 | 43.50 | 43.50 | 55.09 |
| Expert | 58.75 | 39.06 | 45.75 | 46.00 | 43.60 |
| *Static Merging* | | | | | |
| Model Ensemble | 28.00 | 63.16 | 31.25 | 26.50 | 40.30 |
| Model Souping | 44.00 | **79.09** | 43.50 | 43.50 | 55.36 |
| Task Arithmetic | 48.75 | 47.97 | 41.50 | 44.75 | 44.74 |
| Slerp | 44.00 | 79.09 | 43.50 | 43.50 | 55.36 |
| Ties Merging | 51.25 | 75.22 | 43.50 | 48.75 | 55.82 |
| Mixup Merging | 43.50 | 78.41 | 44.50 | **48.25** | 57.05 |
| *Dynamic Merging* | | | | | |
| DaWin | 55.25 | 78.88 | 45.25 | 44.75 | 56.29 |
| $\mathbb{T}^3$ (Ours) | 52.50 | 78.69 | **46.50** | 44.75 | **56.65** |
| $\mathbb{T}^3_{\mathcal{B}}$ (Ours) | **59.25** | 79.09 | 40.50 | 47.75 | 55.78 |

| Retinal OCT → | In-Domain | B2N | Corruptions | | *mean* |
| Methods ↓ | OCTMNIST | OCT | Noise | Digital | mean |
| --- | --- | --- | --- | --- | --- |
| Pretrained | 23.90 | 30.79 | 26.70 | 29.80 | 29.10 |
| Expert | 83.90 | 29.80 | 67.20 | 42.80 | 46.60 |
| *Static Merging* | | | | | |
| Model Ensemble | 25.00 | 19.42 | 27.20 | 25.20 | 23.94 |
| Model Souping | 64.40 | 23.37 | 44.90 | 30.20 | 32.82 |
| Task Arithmetic | 73.60 | 25.34 | 61.50 | 36.20 | 41.01 |
| Slerp | 64.40 | 23.38 | 44.90 | 30.20 | 32.83 |
| Ties Merging | **92.30** | 21.44 | 68.90 | 41.30 | 43.88 |
| Mixup Merging | 22.10 | 11.05 | 63.20 | 25.00 | 33.08 |
| *Dynamic Merging* | | | | | |
| DaWin | 26.00 | **30.78** | 60.30 | 39.90 | 43.66 |
| $\mathbb{T}^3$ (Ours) | 83.50 | 29.40 | 66.80 | 42.40 | 46.20 |
| $\mathbb{T}^3_{\mathcal{B}}$ (Ours) | 83.70 | 29.82 | 67.30 | **42.70** | **46.61** |

baselines across both accuracy and robustness metrics (Figure 6). Table 2 shows that $\mathbb{T}^3_{\mathcal{B}}$ achieves superior or competitive mean performance across modalities: 61.36 (Cell Microscopy) vs. Static Merging's 13.55 and DaWiN's 14.15; 68.94 (Breast Imaging) vs. 66.18 (best baseline); and 46.61 (Retinal OCT) vs. Pretrained Expert's 46.60. Notably, $\mathbb{T}^3_{\mathcal{B}}$ attains near-expert accuracy (98.66 vs. 98.68 on BloodMNIST) while generalizing better to novel distributions. In robustness (Table 7), $\mathbb{T}^3_{\mathcal{B}}$ achieves significantly lower error rates ($mCE$), indicating stronger OOD reliability. For Cell Microscopy, $\mathbb{T}^3_{\mathcal{B}}$ yields 44.42 $mCE$ vs. DaWiN's 99.03 and Static Merging's 99.77. Similarly, in Breast Imaging, $\mathbb{T}^3_{\mathcal{B}}$ attains 68.55 $mCE$, outperforming DaWiN (79.84) and Static Merging (97.91). This stems from our mutual information-based dynamic coefficient $I(x)$, which quantifies model consensus more effectively than DaWiN's entropy ratio $R(x)$. While $R(x)$ conflates confidence with agreement, $I(x)$ explicitly captures disagreements (high JS divergence) even when both models are confident, avoiding erroneous aggregation. Empirical correlations show $I(x)$ better distinguishes conflicting predictions (Figure 3), leading to principled weighting and improved robustness without sacrificing in-domain accuracy.

**Modality Consistency:** Furthermore, entropy ratio-based approaches like DaWiN (Eq. 3) fail to discern whether to prioritize the pretrained or fine-tuned model's predictions. This ambiguity in dependency leads to critical generalization failures, as evidenced by DaWiN's low accuracy (14.15) in Cell Microscopy compared to $\mathbb{T}^3_{\mathcal{B}}$'s significantly higher 61.36. $\mathbb{T}^3$ and $\mathbb{T}^3_{\mathcal{B}}$ demonstrate remarkable consistency across diverse medical imaging modalities, maintaining superior performance in both in-domain and OOD scenarios. This cross-modality robustness stems from the dynamic merging's ability to adaptively balance pretrained and fine-tuned model predictions based on their consensus patterns. While absolute performance varies due to inherent modality-specific challenges, the relative improvement over baselines shows minimal variance, approximately 2-3× performance gain over DaWin across all domains. This consistency extends to robustness metrics where error reduction follows similar patterns regardless of domain shift type. The method's reliance on mutual information rather than raw prediction confidence enables it to transcend modality-specific characteristics, as it focuses on the structural relationship between model outputs rather than the outputs themselves, yielding an approach that generalizes effectively across varied medical imaging contexts.

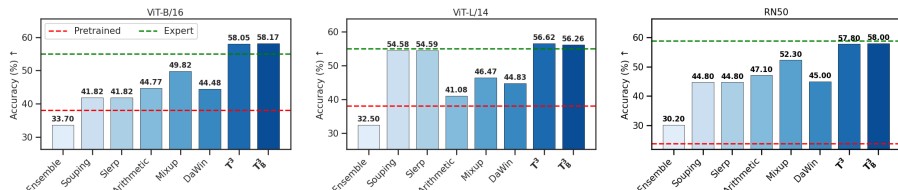

Figure 6: **Mean Top-1 Accuracy** *averaged* **across four medical modalities and different backbones.** Consistent generalization across diverse clinical imaging tasks is crucial in medical settings. Our test-time merging, $\mathbb{T}^3_{(B)}$, outperforms all static and dynamic baselines, including the Expert CLIP, on ViT-B/16 (left), ViT-L/14 (middle), and ResNet-50 (right) backbones. This demonstrates that mutual-information-guided fusion yields more reliable performance across medical modalities than single-model or fixed-weight approaches.

Table 3: **Computation Costs** reporting mean results averaged across four modalities for CLIP ViT-B/16 backbone. Given a pretrained and a expert model, inference cost (I) is measured in forward-passes over the entire test set (with $N$ total samples, grouped into $B$ batches of size BS so that $N = B \times$ BS, where $N >> B$). All reported results are averaged over three runs using different random seeds. $\lambda$ is computed using Eq. 10.

| Methods $\rightarrow$ | Pretrained | Expert | Model Ensemble | Model Souping | Task Arithmetic | Slerp | Mixup Merging | DaWin | No Precompute $\lambda$ | | Precompute $\lambda$ | |
| --- | --- | --- | --- | --- | --- | --- | --- | --- | --- | --- | --- | --- |
| | | | | | | | | | $\mathbb{T}^3$ | $\mathbb{T}^3_{\mathcal{B}}$ | $\mathbb{T}^3$ | $\mathbb{T}^3_{\mathcal{B}}$ |
| OOD *mean* | 38.05 | 55.01 | 33.70 | 41.82 | 44.77 | 41.82 | 49.82 | 44.48 | 58.05 | 58.17 | 58.05 | 58.17 |
| Inference Cost (I) | $\mathcal{O}(1B)$ | $\mathcal{O}(1B)$ | $\mathcal{O}(2B)$ | $\mathcal{O}(1B)$ | $\mathcal{O}(1B)$ | $\mathcal{O}(1B)$ | $\mathcal{O}(1B)$ | $\mathcal{O}(3B)$ | $\mathcal{O}(3N)$ | $\mathcal{O}(3B)$ | $\mathcal{O}(N)$ | $\mathcal{O}(1B)$ |
| Time (seconds) | 41.2 | 41.2 | 81.6 | 41.3 | 41.7 | 41.9 | 41.3 | 124.7 | $>3800$ | 123.5 | $\geq 1260$ | 41.9 |

## 5.2 ANALYSIS

**Backbone Generalization:** $\mathbb{T}^3$ demonstrates remarkable backbone-agnostic performance, consistently outperforming all baselines across both ViT-B/16, ViT-L/14, RN50 architectures. As Figure 6 shows averaged results across domains, $\mathbb{T}^3$ achieves 58.05%, 56.62%, and 57.80% mean accuracy respectively, exceeding experts and competing methods. By leveraging mutual information between model distributions rather than architecture-specific features, $\mathbb{T}^3$ delivers consistent improvements regardless of the underlying network structure, offering a truly generalizable solution for medical imaging applications. Further Ablation studies and Analysis are discussed in Appendix E.

**Computational Costs:** In practical implementation, inference efficiency is paramount for test-time merging solutions. To address this challenge, we implemented our method with the option to precompute interpolation coefficients before deployment, following DaWin Oh et al. (2024). As shown in Table 3, our approach ($\mathbb{T}^3$) achieves superior OOD generalization (58.05%) while maintaining competitive computational efficiency. Without precomputation, the sample-wise merging variant ($\mathbb{T}^3$) requires $\mathcal{O}(3N)$ inference cost due to the additional forward passes needed to compute the Jensen-Shannon divergence between model distributions. However, the batch-wise variant ($\mathbb{T}^3_{\mathcal{B}}$) significantly reduces this to $\mathcal{O}(3B)$, making the cost dependent on batch count rather than sample count. Additional precomputation details are highlighted in Appendix D.

Most impressively, with precomputation, $\mathbb{T}^3$ maintains its performance while reducing inference cost to $\mathcal{O}(3N)$ and $\mathcal{O}(1B)$ for $\mathbb{T}^3$ and $\mathbb{T}^3_{\mathcal{B}}$ repsectively, achieving the same speed as vanilla pretrained and expert models at $\mathcal{O}(1B)$. This efficiency is reflected in the inference times, where precomputed $\mathbb{T}^3_{\mathcal{B}}$ completes processing in just 41.3 seconds, identical to the expert/pretrained models and substantially faster than competing methods like DaWin (124.7 seconds). This computational parity with single models, combined with our superior OOD mean accuracy demonstrates that our approach successfully eliminates the traditional accuracy-efficiency tradeoff in model merging.

## 6 CONCLUSION

In this work, we have proposed $\mathbb{T}^3$, a backpropagation-free, mutual-information-guided framework for dynamic test-time merging of a pretrained generalist and a fine-tuned expert model across diverse medical modalities. By leveraging Jensen-Shannon divergence to measure consensus between their full predictive distributions, our sample-wise ($\mathbb{T}^3$) and batch-wise ($\mathbb{T}^3_{\mathcal{B}}$) variants allocate adaptive interpolation weights that both preserve specialist insights and maintain broad robustness under domain shifts. Empirical results on four challenging medical imaging modalities demonstrate consistently high OOD accuracy and corruption resilience, while matching the inference cost and latency of a single CLIP backbone via batch-wise merging and precomputing the interpolation coefficient. A promising direction would be to extend $\mathbb{T}^3$ to large language models, enabling adaptive model merging across different tasks achieving better test-time scaling.

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

# APPENDIX

This Appendix provides supplementary material for the main paper, "📦 $\mathbb{T}^3$: **Test-Time Model Merging in Vision-Language Models for Zero-Shot Medical Imaging Analysis**". Due to space constraints, extended implementation details and baseline descriptions were omitted from the main text and additional contents. In this Appendix, we include:

## A TERMINOLOGY NOTATIONS

Table 4: Terminology used throughout this paper, with concise descriptions for clarity and ease of reference.

| Term (A-Z) ↓ | Description ↓ |
|---|---|
| Dynamic | Adaptive merging that adjusts weights per sample or batch. |
| Domain | A specific data distribution within a modality (e.g., a particular dataset). |
| Expert Model | The fine-tuned specialist model trained on the target dataset. |
| Coefficient ($\lambda$) | Weight in $[0, 1]$ that blends pretrained and expert parameters. |
| JS Divergence | Jensen–Shannon divergence measuring full-distribution disagreement. |
| KL Divergence | Kullback–Leibler divergence: $\mathrm{KL}(p\|q) = \sum p \log(p/q)$. |
| MediMeta | Standarized database consisting various medical imaging datasets. |
| Modality | Type of data source (e.g., cell microscopy, breast imaging, OCT). |
| MVLM | Medical Vision-Language Model (e.g., CLIP, MedCLIP). |
| OOD | Out-of-Distribution: samples not drawn from the seen distribution. |
| Pretrained Model | The generalist model trained on large, broad-scale data (e.g., CLIP). |
| Softmax | Activation turning logits into a probability distribution. |
| Severity | Intensity level of synthetic corruptions in MedMNIST-C. |
| Static Merging | Weight fusion performed once offline, before any inference. |
| Test-Time | Performing operation during inference, i.e., while being online. |
| Zero-Shot | Inference on a new task without any task-specific training examples. |

## B ADDITIONAL RELATED WORKS

**Test-Time Adaptations:** Existing Test-Time Adaptation (TTA) methods (e.g., TPT, TTL, TDA, SwapPrompt Shu et al. (2022); Imam et al. (2024)) have demonstrated that tailoring adaptation at test time can substantially improve robustness. However, these methods rely on entropy optimization on a per-sample basis, tending to *overfit*, yielding only superficial improvements that fail to translate into true generalization, and can vary from model to model. Such limitations are particularly hazardous in real-world medical applications, where reliable and consistent performance is critical. Since our work combines two models for a downstream task, we omitted these methods as comparative baselines because they involve adapting only a single model during inference.

**VLMs for Medical Imaging:** MVLMs such as MedCLIP Wang et al. (2022b) and BioMedCLIP Zhang et al. (2025) leverage large-scale pretraining on diverse medical image-text pairs, followed by domain-specific fine-tuning to enhance diagnostic accuracy. Due to inherent domain shifts and modality-specific challenges, these models often exhibit degraded performance under distribution shifts, underscoring the need for robust adaptation techniques Radford et al. (2021); Wang et al.

Table 5: Overview of dataset statistics for MediMeta Woerner et al. (2024a) and MedMNIST Chen et al. (2021), covering the common imaging modalities analyzed in this study. #Val/Test represents the number of validation and test samples. Please note that we only evaluated the merging methods on *testset* of each dataset, where *Expert* model is fine-tuned on MedMNIST trainset.

| Modality ↓ | MediMeta | | | MedMNIST / MedMNIST-C | | |
|---|---|---|---|---|---|---|
| | Data Name | #Val/Test | Description | Data Name | #Val/Test | Description |
| Cell Microscopy | PBC | 1,709/3,149 | Blood cells | BloodMNIST | 1,712/3,421 | Blood cells |
| Breast Imaging | Mammo | 214/326 | Calcifications | BreastMNIST | 78/156 | Breast tumors |
| Fundoscopy | Fundus | 640/640 | Eye diseases | RetinaMNIST | 120/400 | Eye diseases |
| Retinal OCT | OCT | 16,694/1,000 | Retinal layers | OCTMNIST | 10,832/1,000 | Retinal layers |

Table 6: Data Sources of Medical Modalities in MedMNIST and MediMeta used in our benchmark Evauation setup, detailing the sources, demographics, and image characteristics of all eight datasets validating their *provenance*.

| MedMNIST | Source ∣ Demographics ∣ Characteristics |
|---|---|
| BloodMNIST | Source: Acevedo et al. (2020)
Demographics: ∼17K peripheral blood cell images from healthy donors across 8 cell types.
Characteristics: RGB microscopy photos, cropped and resized to $28 \times 28$. |
| BreastMNIST | Source: Al-Dhabyani et al. (2020)
Demographics: 780 breast ultrasound images from ∼600 women aged 25–75; labels: normal (133), benign (487), malignant (210).
Characteristics: Grayscale B-mode ultrasound, resized to $28 \times 28$. |
| RetinaMNIST | Source: Liu et al. (2022)
Demographics: 1,600 fundus images labeled by grade (0–4) from screened diabetic patients.
Characteristics: RGB fundus photos, center-cropped and resized to $28 \times 28$. |
| OCTMNIST | Source: Kermany et al. (2018)
Demographics: ∼109K OCT retinal scans across 4 classes (CNV, DME, drusen, normal).
Characteristics: Grayscale OCT B-scans, cropped and resized to $28 \times 28$. |

| MediMeta | Source ∣ Demographics ∣ Characteristics |
|---|---|
| PBC | Source: Acevedo et al. (2020)
Demographics: ∼17K RBC images from healthy donors across 8 classes.
Characteristics: RGB microscopy photos via CellaVision DM96, resized to $224 \times 224$. |
| Mammo | Source: Lee et al. (2017)
Demographics: 3,568 mammography ROIs (calcifications and masses) from screened patients.
Characteristics: Grayscale ROI patches, squared and resized to $224 \times 224$. |
| Fundus | Source: Pachade et al. (2021)
Demographics: 3,200 adult fundus images annotated for 46 ocular conditions by expert clinicians.
Characteristics: RGB fundus photos from three camera types, resized to $224 \times 224$. |
| OCT | Source: Kermany et al. (2018)
Demographics: ∼84K retinal OCT scans across 4 diagnostic classes.
Characteristics: Grayscale OCT B-scans, center-cropped and resized to $224 \times 224$. |

(2022c). Notably, no existing solution addresses model merging in an unsupervised manner for such fine-tuned VLMs, leaving a critical gap in achieving robust generalization in medical imaging.

**Optimization-based Model Merging:** Recently, approaches to model merging include methods performing optimization or some form of unsupervised coefficient learning to blend pretrained and expert weights without access to original training data. AdaMerging Yang et al. (2023) leverages entropy minimization on unlabeled test samples to iteratively refine task- or layer-specific interpolation coefficients, yielding substantial gains in multi-task settings. Ties-Merging Yadav et al. (2023) tackles parameter interference by trimming negligible fine-tuned weights, resolving sign conflicts, and merging only sign-aligned parameters, which enhances robustness across modalities and architectures. Huang et al. (2024) is training-free, but its complex merging approach may create compatibility issues across diverse architectures or limit interpretability. Other similar works Gupta et al. (2020); Wang et al. (2022a) also utilize optimization for merging, underscoring the power of surrogate objectives for adaptive, efficient multi-model integration, but their reliance on labeled data restricts their applicability in test-time or zero-shot settings where supervision is unavailable.

## C  ALGORITHM AND EXTENDED DETAILS OF $\mathbb{T}^3$

---

**Algorithm C. PyTorch Style Pseudocode for Mutual Information-Guided Interpolation**

```
# x : single test sample
# model_pt : pretrained model
# model_ft : finetuned model
# lam_min, lam_max : lower and upper interpolation bounds
# eps : small constant for numerical stability

def T^3(model_pt, model_ft, x, lam_min, lam_max, eps):
    # 1. Compute probability distributions for sample x from both models
    p_pt = torch.softmax(model_pt(x), dim=-1)
    p_ft = torch.softmax(model_ft(x), dim=-1)

    # 2. Compute average predictive distribution
    p_bar = (p_pt + p_ft) / 2.0

    # 3. Compute KL divergence for each model with respect to the average
        distribution
    kl_pt = torch.sum(p_pt * torch.log(p_pt / (p_bar + eps)), dim=-1)
    kl_ft = torch.sum(p_ft * torch.log(p_ft / (p_bar + eps)), dim=-1)

    # 4. Calculate mutual information (MI) as the average KL divergence
    MI = 0.5 * (kl_pt + kl_ft)

    # 5. Map MI to an interpolation coefficient lambda using a sigmoid function
    lam = lam_min + (lam_max - lam_min) * torch.sigmoid(MI)

    # 5.1. Extrapolation for extreme confidence
    H_pt = -torch.sum(p_pt * torch.log(p_pt + eps), dim=-1)
    H_ft = -torch.sum(p_ft * torch.log(p_ft + eps), dim=-1)
    # tau_pt, tau_ft = 0.05, 0.05 : entropy thresholds for pt and ft
    lam = torch.where(
       H_ft < tau_ft,
       torch.clamp(lam + delta, max=1.0),
       torch.where(
          H_pt < tau_pt,
          torch.clamp(lam - delta, min=0.0),
          lam))

    # 6. Merge model parameters using lambda
    merged_state = { key: (1 - lam)*v + lam*model_ft.state_dict()[key]
                  for key, v in model_pt.state_dict().items() }

    # 7. Return Merged model
    return model_pt[merged_state]
```

---

## D  DETAILS ON DATASET AND EXPERIMENTATION

### D.1  DATASET

**Cross-Dataset Benchmark:** To deepen the intuition for our cross-dataset benchmark and its role in model-merging, we emphasize three guiding principles. First, real-world clinical deployment rarely mirrors a single "clean" train-and-test split: each hospital (or imaging center) embodies its own idiosyncratic data distribution, differences in scanner hardware, patient demographics, or acquisition settings, that profoundly affect model performance. By fine-tuning CLIP on MedMNIST's modality-specific split (e.g. BloodMNIST, BreastMNIST) as our "in-domain" expert, we capture this institution-specific baseline.

Second, true robustness demands both semantic generalization (new disease classes, novel image sources) and resilience to low-level artifacts (sensor noise, pixelation from compressions). Our two

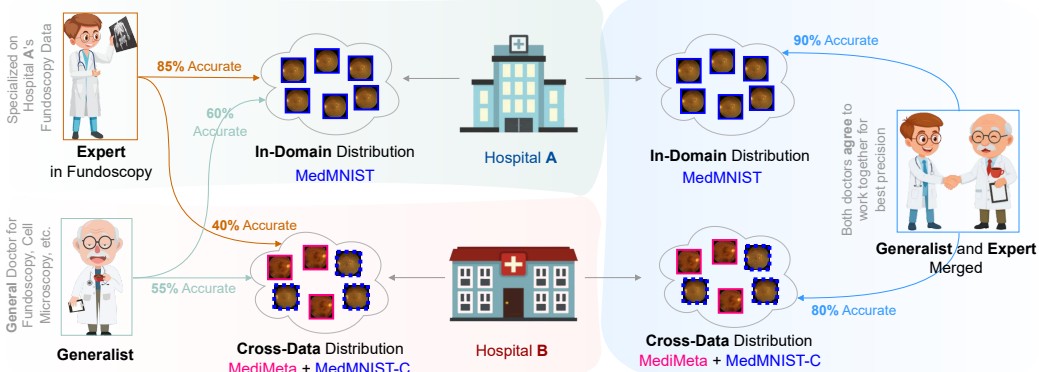

Figure 7: **Use-case Illustrating the Advantage of Model Merging in Real-World Healthcare Settings.** This example demonstrates our cross-data evaluation benchmark applied to fundoscopy classification, though it generalizes consistently across medical modalities. By combining a generalist doctor/model with an expert doctor/model through dynamic merging, they jointly achieve higher precision than either could alone.

> ↪ *Specifically, specialist doctor (Expert) who excels at interpreting scans from Hospital A but struggles with the slightly different protocols at Hospital B. By contrast, a veteran clinician (Generalist) delivers reliable, if unspectacular, readings at both sites. When these two doctors collaborate, letting the specialist guide cases it knows best and deferring to the generalist on unfamiliar scans, or mutually reaching consensus in cases of disagreements, they collectively achieve strong performance across both hospitals' data distributions.*

OOD axes, base-to-novel classification from MediMeta Woerner et al. (2024a) and MedMNIST-C corruptions Di Salvo et al. (2024), systematically stress models along these orthogonal dimensions. Third, by unifying these four modalities (cell microscopy, breast imaging, fundoscopy, retinal OCT) under one protocol, we create a reusable framework that supports fair, head-to-head evaluation of any merging strategy. This design enables practitioners to measure not only overall accuracy but also the interplay between domain shift and artifact severity, producing actionable insights for dynamic model merging in safety-critical applications. Table 5 depicts data statistics for MediMeta and MedMNIST(-C).

Table 6 depicts provenance of datasets used in our cross-evaluation illustrating distribution shifts in our setup. Even when datasets share provenance, modality-specific preprocessing induces a genuine distribution shift: BloodMNIST and MediMeta PBC both stem from same source, yet BloodMNIST uses 28×28 center-cropped, normalized RGB patches while PBC uses 224×224 bicubic-resized, artifact-free scans; aligning with the performance gap observed in Table 2. Likewise, OCTMNIST and MediMeta OCT (both citing same source) differ by center-cropped, normalized patches versus bicubic-resized scans; the Expert attains 83.90% on OCTMNIST but 29.80% on OCT, underscoring preprocessing alone as a major driver of shift.

## D.2   METRICS AND BASELINES

**Evaluation Metrics:** We evaluate every merging methods on both in-domain and OOD data using two complementary metrics. **Top-1 accuracy** (Acc) measures the fraction of correctly predicted labels and thus quantifies overall predictive performance and generalization to novel or corrupted inputs. **Corruption Error** (Err), motivated by ImageNet-C Hendrycks & Dietterich (2019) is defined for any dataset as the ratio of the model's error rate to that of the pretrained CLIP baseline: $\text{Err}_{method} = (1 - \text{Acc}_{method}) / (1 - \text{Acc}_{base})$, where we set *base* as baseline CLIP model consistently across our evaluations. Err therefore captures robustness to distribution shifts: values below 100 indicate a model that degrades less than the CLIP prior when faced with the same perturbations or novel classes, while values above 100 indicate greater sensitivity to distribution shift. By reporting both Acc and Err across all test conditions, we obtain a holistic view of each method's trade-off between accuracy (generalization) and stability (robustness).

**Comparative Baselines:** Our comparison focuses on how to fuse the two given models, a generic CLIP Radford et al. (2021) *pretrained* checkpoint and its *expert* fine-tuned counterpart, using a variety

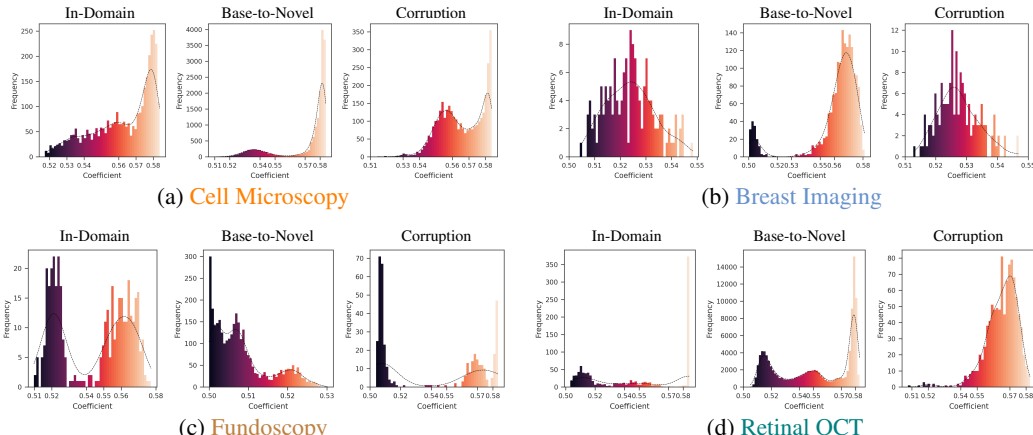

Figure 8: **Histogram of interpolation coefficients** induced by our Mutual Information-based coefficient $\lambda(x)$ (from Eq. 10) between pretrained and expert models. For each modality and under three test settings: In-Domain (data seen to *expert* during fine-tuning), Base-to-Novel (cross-dataset generalization), and Corruption inputs.

of *static merging* strategies. First, we compare traditional **Model Ensemble**, which averages the two models' logits at inference, and **Model Souping** Wortsman et al. (2022a), which weight-space averages their parameters à la WiSE-FT Wortsman et al. (2022b). Next, we test **Task Arithmetic** Ilharco et al. (2022), where the "task vector" (difference between expert and pretrained weights) is added back to the pretrained model, as well as **Slerp** White (2016), a spherical linear interpolation in weight space. We include **Ties Merging** Yadav et al. (2023) which address the problem of interference in merging by trimming, electing, and merging the weights, and finally we include **Mixup Model Merging** Zhou et al. (2025), which inspired by the Mixup data augmentation, performs randomized linear interpolation ratios during model merging. These methods probe whether label-free and backpropagation-free fusions of two base models can match the gains of a more nuanced, per-sample approach.

Beyond static fusions, we benchmark against the recent *dynamic merging*, **DaWin** Oh et al. (2024), which uses each sample's entropy ratio $R(x)$ to choose between the pretrained and expert models on the fly. By contrasting our mutual-information-guided interpolation with DaWin's entropy-based criterion, we isolate the benefit of capturing true inter-model agreement rather than single-model confidence.

### D.3 ADDITIONAL IMPLEMENTATION DETAILS

**Setup:** In our model-merging framework, we maintain two complementary networks with homogenous architecture: a pretrained "generalist" CLIP backbone and a family of "expert" models obtained through supervised fine-tuning on each modality's in-domain training set. Concretely, we start from the off-the-shelf CLIP ViT-B/16 weights and fine-tune separate experts on Cell Microscopy, Breast Imaging, Fundoscopy, and Retinal OCT data for 50 epochs (batch size = 32, learning rate = 1e-5 with AdamW), optimizing cross-entropy loss over ground-truth labels. To avoid overfitting, we apply standard augmentations (random crop, horizontal flip) and early stopping based on validation accuracy. At test time, the generalist provides broad visual-text alignment, while each expert contributes specialized discriminative power, enabling adaptive merging that leverages the strengths of both. This dual-model design underpins our cross-dataset evaluation protocol, where we systematically merge pretrained and expert models across all four modalities to assess OOD generalization and corruption resilience.

**Precomputation of $\lambda(x)$:** Sample-wise merging normally needs three forward passes per input (generalist, expert to get $\lambda(x)$, then merged). We cut this to one pass by pre-computing $\lambda$ offline using Eq. (10) with the JS divergence: (1) Offline scan: run generalist and expert over the eval set in batches, compute $D_{\mathrm{JS}}(x)$ for each sample, and map it to $\lambda(x)$. (2) Cache: store all $\lambda(x)$ values (or per-batch means for $\mathrm{T}_{\mathcal{B}}^3$) to disk. (3) Inference: for each batch, load the pre-computed $\lambda$ (per-sample for $\mathrm{T}^3$, batch mean for $\mathrm{T}_{\mathcal{B}}^3$), form $W_\lambda = (1 - \lambda)W_{\mathrm{gen}} + \lambda W_{\mathrm{exp}}$ once, and run a single forward pass. This keeps predictions identical to the online version while reducing test-time cost to one forward pass per batch and removing extra merging overhead.

Table 7: Comparison of Error rates *Err*, i.e., *Robustness* (↓), for In-Domain, Base-to-Novel, and Corruption settings using CLIP ViT-B/16 on various modalities. mCE indicates the mean corruption error across all shifts. **Bold** = best, underlined = second-best (among merging baselines).

| Cell Microscopy → | In-Domain | B2N | Corruptions | | mCE |
|---|---|---|---|---|---|
| Methods ↓ | BloodMNIST | PBC | Noise | Digital | mean |
| Pretrained | 100.00 | 100.00 | 100.00 | 100.00 | 100.00 |
| Expert | 1.57 | 79.74 | 14.21 | 39.62 | 44.52 |
| *Static Merging* | | | | | |
| Model Ensemble | 101.74 | 101.44 | 100.83 | 97.04 | 99.77 |
| Model Souping | 90.37 | 107.51 | 95.93 | 89.62 | 97.68 |
| Task Arithmetic | 51.67 | 99.14 | 57.21 | 87.40 | 81.25 |
| Slerp | 90.34 | 107.55 | 95.93 | 89.62 | 97.70 |
| Mixup Merging | **1.54** | **79.71** | 81.75 | 37.38 | 66.28 |
| *Dynamic Merging* | | | | | |
| DaWin | 99.15 | 99.95 | 98.75 | 98.40 | 99.03 |
| $\mathbb{T}^3$ (Ours) | 1.74 | 80.35 | **15.74** | 38.68 | 44.92 |
| $\mathbb{T}^3_{\mathcal{B}}$ (Ours) | 1.60 | 79.76 | 16.33 | **37.18** | **44.42** |

| Breast Imaging → | In-Domain | B2N | Corruptions | | mCE |
|---|---|---|---|---|---|
| Methods ↓ | BreastMNIST | Mammo | Noise | Digital | mean |
| Pretrained | 100.00 | 100.00 | 100.00 | 100.00 | 100.00 |
| Expert | 40.63 | 84.43 | 68.17 | 54.76 | 69.12 |
| *Static Merging* | | | | | |
| Model Ensemble | 81.23 | 87.28 | 113.61 | 92.85 | 97.91 |
| Model Souping | 51.55 | 100.00 | 95.43 | 52.39 | 82.60 |
| Task Arithmetic | 73.43 | **81.48** | 120.42 | 55.95 | 85.95 |
| Slerp | 51.55 | 100.00 | 95.43 | 52.39 | 82.60 |
| Mixup Merging | 42.19 | 84.99 | 100.00 | **51.18** | 78.72 |
| *Dynamic Merging* | | | | | |
| DaWin | 70.31 | 100.45 | 79.55 | 59.52 | 79.84 |
| $\mathbb{T}^3$ (Ours) | 40.63 | 84.21 | **68.17** | 55.95 | 69.44 |
| $\mathbb{T}^3_{\mathcal{B}}$ (Ours) | 40.63 | 83.89 | **68.17** | 53.57 | **68.55** |

| Fundoscopy → | In-Domain | B2N | Corruptions | | mCE |
|---|---|---|---|---|---|
| Methods ↓ | RetinaMNIST | Fundus | Noise | Digital | mean |
| Pretrained | 100.00 | 100.00 | 100.00 | 100.00 | 100.00 |
| Expert | 73.01 | 280.57 | 96.02 | 95.58 | 157.39 |
| *Static Merging* | | | | | |
| Model Ensemble | 127.43 | 169.61 | 121.68 | 130.09 | 140.46 |
| Model Souping | 99.12 | 96.27 | 100.00 | 100.00 | 98.76 |
| Task Arithmetic | 90.71 | 239.55 | 103.54 | 97.79 | 146.96 |
| Slerp | 99.12 | 96.27 | 100.00 | 100.00 | 98.76 |
| Mixup Merging | 100.00 | 99.40 | 98.23 | **91.59** | **96.41** |
| *Dynamic Merging* | | | | | |
| DaWin | 79.20 | 97.24 | 96.90 | 97.79 | 97.31 |
| $\mathbb{T}^3$ (Ours) | 84.07 | 98.11 | **94.69** | 97.79 | 96.86 |
| $\mathbb{T}^3_{\mathcal{B}}$ (Ours) | **72.12** | **96.27** | 105.31 | 92.48 | 98.02 |

| Retinal OCT → | In-Domain | B2N | Corruptions | | mCE |
|---|---|---|---|---|---|
| Methods ↓ | OCTMNIST | OCT | Noise | Digital | mean |
| Pretrained | 100.00 | 100.00 | 100.00 | 100.00 | 100.00 |
| Expert | 21.16 | 101.43 | 44.75 | 81.48 | 75.89 |
| *Static Merging* | | | | | |
| Model Ensemble | 98.55 | 116.43 | 99.32 | 106.55 | 107.43 |
| Model Souping | 46.78 | 110.72 | 75.17 | 99.43 | 95.11 |
| Task Arithmetic | 34.69 | 107.87 | 52.52 | 90.88 | 83.76 |
| Slerp | 46.78 | 110.71 | 75.17 | 99.43 | 95.10 |
| Mixup Merging | 102.37 | 128.52 | 50.20 | 106.84 | 95.19 |
| *Dynamic Merging* | | | | | |
| DaWin | 97.24 | **100.01** | 54.16 | 85.61 | 79.93 |
| $\mathbb{T}^3$ (Ours) | 21.68 | 102.01 | 45.29 | 82.05 | 76.45 |
| $\mathbb{T}^3_{\mathcal{B}}$ (Ours) | **21.42** | 101.40 | **44.61** | 81.62 | **75.88** |

(a) Top-1 Accuracy

(b) Error (Err)

Figure 9: **Effect of extrapolation factor $\delta$ on generalization**. **mean** denotes averaged results across four modalities Incorporating $\delta = 0.5$ consistently improves both accuracy (a) and robustness (b), with mean accuracy increasing by up to 0.40% while reducing mCE by up to 1.09%. Extended ablation on $\delta$ in Appendix E.

# E    EXTENDED RESULTS AND ABLATIONS

## E.1    RESULTS

**Robustness Results:** Across all four modalities, Cell Microscopy, Breast Imaging, Fundoscopy, and Retinal OCT, our adaptive merging methods ($\mathbb{T}^3$ and $\mathbb{T}^3_{\mathcal{B}}$) achieve the lowest mean corruption error (mCE) in Table 7, outperforming static baselines (Mixup, Task Arithmetic) and prior dynamic schemes (DaWin). For instance, $\mathbb{T}^3_{\mathcal{B}}$ reduces Cell Microscopy's average OOD error to 44.92%, versus 99.03% for DaWin and 100% for the pretrained model. Here, *mCE*, computed as the average normalized error across multiple corruption types, directly measures model robustness to common image degradations. *Crucially, true generalization in our medical setting requires both high in-domain accuracy and low mCE under corruption and cross-datasets*, and $\mathbb{T}^3$ excels on both fronts. As Figure 8 reveals, $\mathbb{T}^3$ adaptively sets its interpolation coefficient $\lambda(x)$ towards 1.0 for benign in-domain samples, leveraging expert knowledge, and shifts toward 0.0 when novel classes or severe corruptions appear, falling back on the pretrained model's broader resilience. This tight correlation between the per-sample coefficient distribution and the observed drop in mCE demonstrates that our dynamic, JS-guided merging is the key driver of enhanced robustness and overall generalization across diverse distribution shifts.

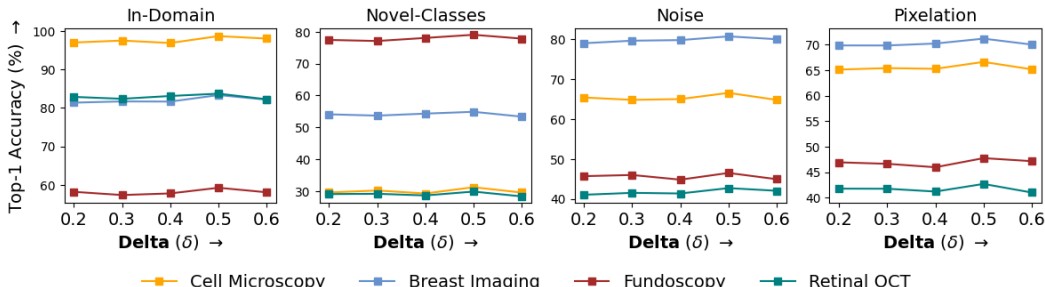

Figure 10: **Delta $\delta$ ablation over four test conditions** (In-Domain, Novel-Classes, Noise, Pixelation) shows Top-1 accuracy for each modality as the extrapolation factor $\delta$ varies from 0.2 to 0.6. All of the reported results are with $\delta = 0.5$ throughout.

Table 8: Ablation across Standard Pretrained *CLIP* vs medical pretrained *BioMedCLIP* bases with ViT-B/16 architectures for Breast Imaging Modality. **Bold** highlights best performance, while underlined denotes second-best performance.

| Methods ↓ | Base ↓ | In-Domain | B2N | Noise | Digital | *mean* ↓ |
|---|---|---|---|---|---|---|
| Pretrained | CLIP | 58.97 | 46.23 | 71.79 | 46.15 | 54.72 |
| | BioMedCLIP | 65.10 | 50.50 | 75.20 | 53.45 | 61.58 |
| Expert | CLIP | 83.33 | 54.60 | 80.77 | 70.51 | 68.63 |
| | BioMedCLIP | 89.15 | 61.37 | 86.42 | 75.18 | 72.23 |
| | *Static Merging* | | | | | |
| Ensemble | CLIP | 66.67 | 53.07 | 67.95 | 50.00 | 57.01 |
| | BioMedCLIP | 72.20 | 57.46 | 72.50 | 54.68 | 61.46 |
| ModelSouping | CLIP | 78.85 | 46.23 | 73.08 | 71.79 | 63.70 |
| | BioMedCLIP | 82.33 | 49.55 | 77.22 | 74.11 | 66.70 |
| TaskArithmetic | CLIP | 69.97 | 56.19 | 66.03 | 69.87 | 64.03 |
| | BioMedCLIP | 75.12 | 63.41 | 71.33 | 72.29 | 68.58 |
| Slerp | CLIP | 78.85 | 46.23 | 73.08 | 71.79 | 63.70 |
| | BioMedCLIP | 81.28 | 48.47 | 75.19 | 74.40 | 65.62 |
| MixupMerging | CLIP | 82.69 | 54.30 | 71.79 | 72.44 | 66.18 |
| | BioMedCLIP | 88.15 | 59.05 | 77.92 | 74.60 | 73.19 |
| TiesMerging | CLIP | 78.21 | 54.54 | 76.28 | 74.36 | 68.39 |
| | BioMedCLIP | 84.50 | 59.35 | 82.18 | 78.66 | 73.40 |
| | *Dynamic Merging* | | | | | |
| DaWin | CLIP | 71.15 | 45.99 | 77.56 | 67.95 | 63.83 |
| | BioMedCLIP | 74.15 | 50.47 | 80.39 | 71.28 | 67.38 |
| $\mathbb{T}^3$ (Ours) | CLIP | 83.33 | 54.72 | 80.77 | 69.87 | 68.45 |
| | BioMedCLIP | 90.24 | 60.13 | 86.92 | 74.50 | 73.85 |
| $\mathbb{T}^3_{\mathcal{B}}$ (Ours) | CLIP | 83.33 | 54.89 | 80.77 | 71.15 | **68.94** |
| | BioMedCLIP | 91.02 | 61.47 | 87.35 | 75.81 | **74.88** |

## E.2 ABLATIONS

**Effect of Extrapolation Factor $\delta$:** The extrapolation factor $\delta$ provides incremental refinement to $\mathbb{T}^3$, functioning primarily as a complement that addresses edge cases of extreme model confidence. As Figure 9 demonstrates, while the incorporation of $\delta$ yields slight improvements (0.40% mean accuracy increase and 1.09% mCE reduction), the fundamental performance gains of $\mathbb{T}^3$ derive predominantly from its Jensen-Shannon divergence approach. The extrapolation mechanism serves as a confidence-calibrating supplement that selectively adjusts merging weights only when entropy falls below a critical threshold $\tau$ as in Eq. 11, effectively mimicking clinical decision-making where unusually definitive signals receive heightened consideration.

We swept the extrapolation coefficient $\delta$ to ablate its impact in our $\mathbb{T}^3$ merging framework as in Figure 10. Although $\delta$ plays only a minor role, nudging the interpolation weight when one model's entropy is exceptionally low, it prevents both over- and under-reliance on a single expert under extreme confidence. Across in-domain, novel-class, noise, and pixelation shifts, Top-1 accuracy fluctuates by at most a few percentage points, confirming that $\delta$ stabilizes performance rather than destabilizing it.

Notably, $\delta = 0.5$ consistently delivers the best or tied-best accuracy, outperforming smaller values ($\delta = 0.2, 0.3$) that under-adjust and larger values ($\delta = 0.6$) that over-correct. A mid-range $\delta$ of 0.5 thus strikes the ideal balance, sufficiently amplifying confidence when warranted, yet avoiding excessive weight swings, yielding robust gains across all modalities and distributional conditions.

**Effect of Different Base Models:** "Generalist" refers to any broad pretrained VLM (e.g., CLIP, BioMedCLIP); "Expert" is that model fine-tuned on in-domain data. We used standard CLIP in all our aforementioned experiments to illustrate a conservative scenario. In Table 8, we show that substituting CLIP with BioMedCLIP as the base model further boosts accuracy all merging setups, confirming generalizability as well as that more medical-centric VLMs only strengthen $\mathbb{T}^3$. Replacing CLIP with BioMedCLIP increases the mean accuracy, with gains ranging from +1.92 (Slerp) to +7.01 (MixupMerging), and improvements of +5.40 for $\mathbb{T}^3$ and +5.94 for $\mathbb{T}^3{}_{\mathcal{B}}$. Benefits extend beyond in-domain accuracy: for the non-fine-tuned Pretrained baseline, B2N improves by +4.27 and Digital by +7.30, while $\mathbb{T}^3{}_{\mathcal{B}}$ advances from $54.89 \rightarrow 61.47$ on B2N and $71.15 \rightarrow 75.81$ on Digital, indicating stronger base-to-novel transfer and corruption robustness from the medical-centric backbone.

Static mergers all improve: Ensemble (+4.45), ModelSouping (+3.00), TaskArithmetic (+4.55), Slerp (+1.92), TiesMerging (+5.01), MixupMerging (+7.01), and the dynamic DaWin also gains (+3.55 mean), showing that backbone choice is complementary to the merging algorithm. The Expert baseline rises from $68.63\% \rightarrow 72.23\%$, lifting the attainable ceiling for all downstream mergers. Overall, medical-domain pretraining consistently *amplifies* performance across in-domain, base-to-novel, and corruption settings, and further strengthens $\mathbb{T}^3$ under identical training budgets. This finding would encourage future works to build on our work with even more specialized model combinations.

Table 9: Comparison of basis for computing interpolation coefficients for test-time merging: Entropy-Ratio (DaWiN Eq. 3), Confidence-Ratio (CR), and JS-Divergence (Ours $\mathbb{T}^3$ Eq. 5); based on uncertainty, prediction confidence, and distributional divergence, respectively. Here Confidence ratio$(x) = \max_i \left( p_{\mathrm{ft}}^i(x) \right) / \left[ \max_i \left( p_{\mathrm{pt}}^i(x) \right) + \max_i \left( p_{\mathrm{ft}}^i(x) \right) \right]$.

| Modality ↓ | Methods ↓ | In-Domain | B2N | Noise | Digital | mean ↓ |
|---|---|---|---|---|---|---|
| Cell Microscopy | Entropy ratio | 16.87 | 13.77 | 17.10 | 11.58 | 14.15 |
| | Confidence ratio | 17.37 | 13.27 | 17.90 | 10.78 | 13.98 |
| | JS-Divergence | 98.54 | 30.68 | 86.79 | 65.24 | **60.90** |
| Breast Imaging | Entropy ratio | 71.15 | 45.99 | 77.56 | 67.95 | 63.83 |
| | Confidence ratio | 70.20 | 46.50 | 78.00 | 67.00 | 63.17 |
| | JS-Divergence | 83.33 | 54.72 | 80.77 | 69.87 | **68.45** |
| Fundoscopy | Entropy ratio | 55.25 | 78.88 | 45.25 | 44.75 | 56.29 |
| | Confidence ratio | 55.75 | 79.50 | 44.00 | 45.50 | 56.33 |
| | JS-Divergence | 52.50 | 78.69 | 46.50 | 44.75 | **56.65** |
| Retinal OCT | Entropy ratio | 26.00 | 30.78 | 60.30 | 39.90 | 43.66 |
| | Confidence ratio | 26.50 | 30.00 | 59.50 | 40.50 | 43.33 |
| | JS-Divergence | 83.50 | 29.40 | 66.80 | 42.40 | **46.20** |

**Ablating Different Interpolating Coefficients:** We find that the Jensen-Shannon (JS) divergence $I(x)$ is agreement-aware: it distinguishes cases where both models are confident but disagree from cases where they are confidently aligned, unlike entropy- or confidence-ratio heuristics that conflate these regimes. In Table 9, using $I(x)$ as the interpolation coefficient attains the best mean across all four modalities and both in-domain and OOD axes: Cell Microscopy 60.90 vs. 14.15/13.98 (entropy/confidence), Breast Imaging 68.45 vs. 63.83/63.17, Fundoscopy 56.65 vs. 56.29/56.33, and Retinal OCT 46.20 vs. 43.66/43.33, with consistent improvements across B2N and corruption columns. Because $I(x)$ is symmetric and compares full predictive distributions, it increases weight when experts truly agree and down-weights confident disagreements, yielding more reliable test-time interpolation under distribution shift. To our knowledge, this is the first use of JS divergence for merging to explicitly separate consensus from disagreement, and the gains indicate it is a principled replacement for entropy ratio.

**Effect of Batch Size:** Across six batch sizes, $\mathbb{T}^3$ remains stable with modest variation per modality, indicating low sensitivity to mini-batch choice and consistent robustness under distribution shift. We set `BS = 1` for $\mathbb{T}^3$ to preserve strict per-sample test-time interpolation (no cross-sample coupling)

Table 10: Ablation of batch size (BS) for $\mathbb{T}^3$ across four modalities on CLIP ViT-B/16. Columns report *mean* accuracy across Distribution shifts where Distribution shifts $\epsilon$ {Base-to-Novel (B2N), Corruption settings}. BS $= 1$ corresponds to $\mathbb{T}^3$ and BS $= 32$ to $\mathbb{T}^3_B$.

| Modality ↓ | BS = 1 ($\mathbb{T}^3$) | BS = 16 | BS = 32 ($\mathbb{T}^3_B$) | BS = 64 | BS = 128 | BS = 256 |
|---|---|---|---|---|---|---|
| Cell Microscopy | 60.90 | 61.30 | 61.36 | 60.85 | 61.40 | 60.20 |
| Breast Imaging | 68.45 | 69.10 | 68.94 | 67.90 | 69.00 | 67.50 |
| Fundoscopy | 56.65 | 57.00 | 55.78 | 56.10 | 57.50 | 55.80 |
| Retinal OCT | 46.20 | 46.80 | 46.61 | 45.50 | 46.90 | 45.30 |

and to minimize latency/memory for on-device use, while BS $= 32$ for $\mathbb{T}^3_B$ amortizes coefficient estimation over a small mini-batch for added stability and GPU throughput, staying near the best accuracy across modalities without incurring large memory costs.

## F  LIMITATIONS AND FUTURE DIRECTION

**Limitations.** While effective for medical image classification, $\mathbb{T}^3$ has several technical constraints. It relies on Jensen-Shannon divergence to calibrate interpolation weights between the specialist and generalist outputs, which can be sensitive when one model's prediction distribution is sharply peaked. In practice $\mathbb{T}^3$ adds a confidence threshold ($\tau$) and extrapolation factor ($\delta$) to handle such cases, but these heuristics require careful tuning and may still fail under extreme model overconfidence, potentially leading to unstable blending in other settings. Moreover, $\mathbb{T}^3$ assumes the availability of both a fine-tuned domain expert and a broad pretrained model, a luxury not always present in every healthcare deployment. Finally, all experiments are on zero-shot classification across four medical modalities; the framework's efficacy on other vision-language tasks (e.g. segmentation, detection, captioning) or non-imaging domains remains untested.

**Future Directions:** Addressing these limitations opens several precise research avenues. One direction is to develop alternative or learned agreement metrics (beyond JS divergence) that are robust to confident outputs, perhaps by calibrating uncertainty or using auxiliary models. Extending $\mathbb{T}^3$ to other task types is also imperative: for example, merging models for image segmentation, radiology report generation, or visual question answering on natural and downstream tasks would also confirm $\mathbb{T}^3$'s generality. Integrating with large language models and handling tasks like captioning or speech recognition could be promising next steps. These efforts would rigorously expand $\mathbb{T}^3$'s applicability and robustness beyond its current medical classification setting.

## G  LLM USAGE

We confirm that LLM was used to assist with writing refinement (grammar, wording, and clarity) only. All ideas, analyses, and conclusions are the authors' own.

