# OpenReview forum: "T³: Test-Time Model Merging in VLMs for Zero-Shot Medical Imaging Analysis"
_ICLR.cc/2026/Conference — ICLR 2026 Conference Withdrawn Submission_

### Official Review · Reviewer_6j9S · 2025-10-30

**Soundness:** 3
**Presentation:** 3
**Contribution:** 3
**Rating:** 6
**Confidence:** 4

**Summary:**

The paper introduces a Test-Time Task adaptive model merging framework (T3) for medical vision-language models (MVLMs). The challenge in medical imaging is balancing the performance between generalist models, which are robust but not specialized, and expert models, which perform well in their trained domain but struggle with out-of-distribution (OOD) data. T3 dynamically merges these two models based on their predictive agreement, using Jensen-Shannon divergence to calculate per-sample interpolation coefficients. This method adapts efficiently at test time without the need for backpropagation, addressing various medical imaging tasks under conditions like distribution shifts and corruption.

**Strengths:**

Strengths:
1.The test-time adaptive merging of pretrained and expert models is novel and addresses real-world challenges in medical imaging, where different datasets and conditions can greatly vary.
2.The backpropagation-free nature of the method makes it computationally efficient, which is crucial for medical applications that require quick and reliable inferences.
3.The method outperforms several baselines across multiple medical modalities, demonstrating its robustness and generalization capabilities.
4.The approach works across various imaging modalities, making it applicable in diverse medical settings.
5.The manuscript details the experimental setup and provides clear pseudocode for the T3 algorithm, which should be beneficial for practitioners looking to replicate or build upon this work.

**Weaknesses:**

Weaknesses:
1.While the use of Jensen-Shannon (JS) divergence to guide model merging is well-motivated empirically, the manuscript could benefit from a more formal theoretical justification of why this divergence is the most suitable metric for this task, especially in comparison to other divergences like Kullback-Leibler divergence (KL) or Total Variation (TV). An in-depth theoretical analysis of the advantages and limitations of JS divergence in the context of model merging could provide a more robust foundation for the proposed method.
2.The manuscript demonstrates excellent results across multiple medical modalities. However, it would be insightful to see how T3 performs in the context of more diverse real-world scenarios, such as multi-modal inputs (e.g., combining different imaging modalities such as CT scans and MRI) or more heterogeneous data sources (e.g., cross-institutional data).
3.The authors briefly discuss how the interpolation coefficient λ(x) adapts based on mutual information. However, the interpretability of this process could be expanded. For instance, providing visualizations of how λ(x) varies across different input samples (such as under various corruption types or novel class scenarios) could help readers better understand how the model makes its adaptive decisions.
4.The manuscript compares T3 to several static and dynamic merging methods, such as Model Ensemble, Task Arithmetic, and DaWin. While these comparisons are comprehensive, it would be helpful to also compare T3 against methods that explicitly focus on zero-shot or few-shot medical image classification, such as transfer learning-based approaches or methods that utilize large language models like MedCLIP[1].
5.It would be useful to also consider additional evaluation metrics, such as F1-score, area under the curve (AUC), or confusion matrices, to give a more complete picture of the method's performance, especially when dealing with imbalanced datasets or novel classes.
6.The method shows promise under typical domain shifts and data corruptions, but medical imaging data can be highly variable, and edge cases (e.g., images with extreme noise or artifacts, or rare diseases) might not be well-represented in the current experiments.

**Questions:**

Questions:
1.Why refer to JS divergence as “mutual information” (Eq. 5, Section 3.2)? This is technically incorrect—mutual information is between random variables, not two distributions over the same variable. Please explain.

2.Why exclude TTA methods like TPT[2] or CoOp-based[3] adaptation as baselines? Even if they adapt a single model, they are strong zero-shot competitors in OOD settings. A comparison would better position T3’s value.
3.The paper claims “zero-shot medical imaging analysis,” but the expert model is fine-tuned on labeled in-domain data. Please clarify that “zero-shot” refers only to test-time inference without task-specific adaptation, not the entire pipeline.
4.The paper primarily focuses on classification tasks. How would the approach perform in other medical imaging tasks such as segmentation or detection? Are there plans to extend the framework to such tasks?
5.While T3 is effective when merging pretrained and expert models, how would it perform in scenarios where pretrained models may not be available or where expert models are extremely specialized?

[1]Wang Z, Wu Z, Agarwal D, et al. Medclip: Contrastive learning from unpaired medical images and text[C]//Proceedings of the Conference on Empirical Methods in Natural Language Processing. Conference on Empirical Methods in Natural Language Processing. 2022, 2022: 3876.
[2]Shu Manli, Nie Weili, Huang De-An, Yu Zhiding, Goldstein Tom, Anandkumar Anima, and Xiao Chaowei. Testtime prompt tuning for zero-shot generalization in visionlanguage models. In NeurIPS, 2022.
[3]Kaiyang Zhou, Jingkang Yang, Chen Change Loy, and Ziwei Liu. Learning to prompt for vision-language models. International Journal of Computer Vision (IJCV), 2022.

---

### Official Review · Reviewer_omq4 · 2025-10-30

**Soundness:** 1
**Presentation:** 2
**Contribution:** 1
**Rating:** 2
**Confidence:** 3

**Summary:**

The paper addresses the challenge of combining generalist, pretrained Vision-Language Models (VLMs) with specialized, fine-tuned "expert" models for medical imaging analysis. The authors propose $\mathbb{T^3}$, a Test-Time Task-adaptive merging framework that is backpropagation-free. The core of the method is a novel interpolation coefficient, λ(x), computed per-sample (or per-batch) based on the Jensen-Shannon (JS) divergence between the output distributions of the two models. The framework also introduces a new benchmark for model merging in medical imaging, upon which the authors claim their method achieves state-of-the-art (SOTA) results in accuracy and robustness.

**Strengths:**

**1. Test-Time Practicality:** The proposed $\mathbb{T^3}$ framework is designed to be backpropagation-free. This makes it suitable for test-time adaptation, as it doesn't require optimization, gradient computation, or access to training labels during inference.

**2. Novelty of Mechanism:** The use of Jensen-Shannon divergence as the mechanism for calculating the interpolation coefficient based on output distribution agreement is a novel idea in the model merging space.

**Weaknesses:**

**1. Misleading Tables:** The results in Table 2 are misrepresented, with incorrect bolding favoring the authors' method over superior baselines (e.g., Fundoscopy mean accuracy).

**2. Misleading Complexity Analysis:** The O(B) vs. O(N) claim is fundamentally incorrect. Both methods are O(N) as B=N/BS. This shows a lack of rigor in the theoretical analysis.

**3. Poor Evaluation Strategy:** Creating a new benchmark and claiming SOTA on it is poor scientific practice. The paper fails to demonstrate its method's value on any existing, standardized model-merging benchmark.

**4. Narrow Application:** The paper's contribution is not shown to be general. It is only tested on medical images, making it a poor fit for ICLR.

**5. Unexplained Computational Cost:** The 3800s inference time for the $\mathbb{T^3}$ method in Table 3 is anomalous and unexplained. It suggests the described cost of 3 forward passes is either wrong or the implementation is extremely inefficient, making it non-competitive with methods like DaWin (124.7s).

**6. Weak Theoretical Justification:** The merging logic seems to function as a regularization technique, rather than an optimal selection mechanism. By defaulting to the generalist when predictions tend to align (low JS divergence), its primary effect is to pull the output toward the generalist in a scenario offering limited apparent gain.

**7. Excessive use of analogies:** There are lots of analogies in the paper between two physicians.  There are figures and boxes throughout the body and appendix. I suggest removing everything regarding those analogies from the body of the paper because they are loose representations of the proposed architecture that do not beolong in a scientific paper. If the authors want this loose explanation in the paper I strongly suggest for it to be only in an appendix.

**Questions:**

1. Can you please justify the claim that the batch-wise method has O(B) complexity while the sample-wise has O(N)? Do you agree that since B=N/BS, both methods are asymptotically linear, O(N)?

2. Can you please explain the 3800s inference time for $\mathbb{T^3}$ in Table 3? Why is it over 30 times slower than DaWin (124.7s), when both are described as requiring 3 forward passes?

3. Why use a model merging technique when most of the results where almost equal to the specialist model?

4. Since the method has nothing that is specific to medical imaging in its architecture, why not test in other domains?

---

### Official Review · Reviewer_GBht · 2025-10-31

**Soundness:** 3
**Presentation:** 3
**Contribution:** 3
**Rating:** 4
**Confidence:** 3

**Summary:**

This work presents a backprop-free method for interpolating between specialist and generalist MVLM models in the medical data setting. To motivate the work and evaluate the method, they construct a benchmark and perform extensive experiments across datasets and comparing against many related work.

**Strengths:**

Motivation is very clear that unlike some synthetic natural image tasks, medical data may have varying types of distribution shift within a modality that require finer-grained interpolation between MVLM models. Can work with single inputs or batches of inputs, which gives more controllability in how interpolation happens.

Intuition for the construction of the method is also clearly discussed and figures are well formatted to explain.

The method itself, based on JSD metric, is also very simple and interpretable, but shows strong positive results against static merging techniques. Great to see lots of comparison results done to many related work.

**Weaknesses:**

Unclear what happens if the specialist and generalist models have radically different architectures, since the method uses a simple linear interpolation between parameters. Seems like a huge downside of using this method in practical applications, despite the strong results.

Need error bars to show if this method of dynamic merging is indeed better than the related work DaWin across all modalities.

Typo line 52 "decisions to reach to an"

**Questions:**

Is there intuition for why such a simple linear interpolation would be sufficient for getting combined model weights?

Is the assumption that the two models will have same architectures? Is this a reasonable assumption in practice? Why or why not? To me, this doesn't seem like the most realistic assumption given diversity of models, pace of development, etc.

---

### Official Review · Reviewer_VAaC · 2025-11-08

**Soundness:** 2
**Presentation:** 2
**Contribution:** 2
**Rating:** 2
**Confidence:** 4

**Summary:**

This paper presents a backpropagation-free framework for dynamically merging pretrained and fine-tuned medical vision–language models (MVLMs) to improve zero-shot robustness across diverse imaging modalities. The authors identify a gap between generalist pretrained models (robust but insensitive to modality nuances) and domain-expert fine-tuned models (accurate but overfitted). To bridge this, T3 introduces a per-sample interpolation coefficient derived from the Jensen–Shannon divergence between model output distributions, guiding adaptive parameter fusion at test time. A batch-wise variant averages interpolation weights across samples to reduce computational overhead. The paper further proposes a standardized cross-modality benchmark across MedMNIST, MediMeta, and MedMNIST-C for assessing model-merging methods under in-domain, base-to-novel, and corrupted conditions. Empirical results show that it consistently improves accuracy and while maintaining inference costs comparable to single-model baselines.

**Strengths:**

- The use of mutual information for per-sample adaptive interpolation offers a sound and interpretable improvement over entropy-based schemes.
- Four imaging modalities, multiple OOD shifts, and comparisons to static/dynamic baselines provide convincing empirical support.

**Weaknesses:**

- Theoretical depth is limited: the paper lacks a formal justification for why JS divergence optimally balances confidence and disagreement beyond empirical correlation plots.
- Some baseline comparisons (e.g., to modern multi-adapter or PEFT-based methods) are missing, which could contextualize the relative merit of merging.
- The benchmark’s clinical realism could be enhanced with higher-resolution or multimodal (text + image) tasks.
- Writing style, though clear, is verbose and occasionally repetitive; streamlining could improve focus.
- No exploration of failure cases, e.g., when both models are wrong but confident, despite being central to the JS-based rationale.

**Questions:**

- How sensitive is performance to the λ bounds (λₘᵢₙ, λₘₐₓ) and extrapolation factor δ? Have you tested adaptive scaling per modality?
- Would using layer-wise or token-level mutual information improve merging granularity beyond global JS divergence?
- How does T3 perform when merging more than two experts (e.g., multi-institution specialists)?
- Could the authors release the benchmark splits and code to facilitate fair comparison and adoption?
- How does the approach handle cases where both models are confidently incorrect (high agreement but low accuracy)?

**Details Of Ethics Concerns:**

No ethics concerns.

---

### Note · Authors · 2025-11-12

I have read and agree with the venue's withdrawal policy on behalf of myself and my co-authors.